# Ubiquitination in T-Cell Activation and Checkpoint Inhibition: New Avenues for Targeted Cancer Immunotherapy

**DOI:** 10.3390/ijms221910800

**Published:** 2021-10-06

**Authors:** Shubhangi Gavali, Jianing Liu, Xinyi Li, Magdalena Paolino

**Affiliations:** Center for Molecular Medicine, Department of Medicine Solna, Karolinska Institutet, Karolinska University Hospital Solna, 17176 Solna, Sweden; shubhangi.gavali@ki.se (S.G.); jianing.liu@ki.se (J.L.); xinyi.li@ki.se (X.L.)

**Keywords:** ubiquitination, T cells, checkpoint inhibition, cancer immunotherapy, deubiquitinases, E3 ligases

## Abstract

The advent of T-cell-based immunotherapy has remarkably transformed cancer patient treatment. Despite their success, the currently approved immunotherapeutic protocols still encounter limitations, cause toxicity, and give disparate patient outcomes. Thus, a deeper understanding of the molecular mechanisms of T-cell activation and inhibition is much needed to rationally expand targets and possibilities to improve immunotherapies. Protein ubiquitination downstream of immune signaling pathways is essential to fine-tune virtually all immune responses, in particular, the positive and negative regulation of T-cell activation. Numerous studies have demonstrated that deregulation of ubiquitin-dependent pathways can significantly alter T-cell activation and enhance antitumor responses. Consequently, researchers in academia and industry are actively developing technologies to selectively exploit ubiquitin-related enzymes for cancer therapeutics. In this review, we discuss the molecular and functional roles of ubiquitination in key T-cell activation and checkpoint inhibitory pathways to highlight the vast possibilities that targeting ubiquitination offers for advancing T-cell-based immunotherapies.

## 1. Introduction

For T cells to become activated upon encountering an antigen, a series of quantitative and qualitative signals have to be received by receptors at the cell surface and successfully integrate with numerous intracellular signaling cascades that eventually dictate an appropriate immune response [1]. Once activated, diverse inhibitory mechanisms are triggered to tune the T cells down over time. These essential, yet complex, regulatory mechanisms prevent autoimmune reactivity and exacerbated immune response, that could be detrimental, and permit tissue repair [2].

Using the same core activating and inhibitory pathways, T cells can also detect and respond to malignant cells. During tumorigenesis, cancer cells express neoantigens or altered-self antigens that can trigger innate and adaptive immune responses [3,4]. However, cancer cells can escape immune surveillance and anti-tumor immune responses by exploiting, in their favor, central mechanisms of immune tolerance and T-cell inhibition. They can, for instance, acquire new mutations that change or mask antigens, down-regulate antigen presentation at their plasma membrane, create an immunosuppressive tumor microenvironment, as well as suppress the immune response by activating inhibitory pathways in the T cells [5,6,7].

Immunotherapies that activate the patient’s own immune system, particularly cytotoxic T cells, for stronger, specific, and lasting anti-tumor immune responses have modernized cancer therapy, complementing the more traditional protocols primarily targeting the tumorigenic cells. Although their clinical outcomes are in general very encouraging, they have failed to provide universal care and can have toxicities associated, such as cytokine release syndrome, inflammation of diverse organs, including the nervous system, the liver, the skin and the intestines, as well as diverse cytopenias, specially neutropenia [8,9,10,11]. To provide a new molecular basis for rationally developing alternative and more efficient treatments, it is therefore of paramount importance that we achieve a deeper molecular understanding of the regulatory mechanisms of T-cell activation and inhibition.

Post-translational protein modifications are indispensable to regulate the numerous and complex transduction events involved in T-cell signaling [12,13]. The modification of proteins with ubiquitin controls virtually all immune responses [14,15]. Specifically in anti-tumor immunity, ubiquitination exerts a myriad of cellular and molecular roles, by controlling the function, localization, and stability of proteins [16]. The roles of ubiquitination in the anti-tumor responses of innate immune cells, B cells, Tregs, and tumor cells have been vastly reviewed elsewhere [17,18]. In this review, we will focus on revising the molecular and functional roles of ubiquitinating and deubiquitinating enzymes in key pathways of T-cell activation and inhibition that can offer alternatives for enhancing T-cell anti-tumor responses in cancer immunotherapy.

## 2. Protein Ubiquitination

Ubiquitination is a post-translational modification by which one, but often a chain of, ubiquitin, a small 76-amino acid protein, is covalently attached to a lysine residue within a substrate protein. Ubiquitination impacts protein function through various means, such as by affecting protein stability, turnover, cellular localization, and inducing conformational changes that affect interaction with other proteins [19]. The process occurs in a stepwise manner, with several enzymes sequentially taking part (Figure 1a). The process is initiated by the E1 ubiquitin-activating enzymes, which activates, using ATP, and then transfers the ubiquitin monomer to the E2 ubiquitin-conjugating enzyme. The ubiquitin is finally transferred from the E2 to the substrate guided by the actions of the E3 ubiquitin-ligating enzymes [20]. There are three classes of E3 ligases; the RING (really interesting new gene) type, the HECT (homologous to the E6AP carboxyl terminus) type, and the RBR (RING-IN-Between-RING)-type, which differ in domains, catalytic mechanisms, and number of protein subunits [21,22,23]. Typically, the last amino acid in the ubiquitin, glycine 76, is linked via an isopeptide bond to the NH2 group of an internal lysine residues in the substrates [20]. More recently it was revealed that E2 and E3 enzymes can also link ubiquitin to the N-terminal amines [24], the sulfhydryl group on cysteines [25], or the hydroxyl group on serines and threonines [26,27], forming instead, peptide, thioester, and hydroxyester bonds respectively. These non-conventional types of ubiquitin bonds have been implicated in diverse cellular responses, especially in endocytosis and endoplasmic reticulum-associated degradation [28]. Despite being thermodynamically less stable than lysine-mediated isopeptide bonds, they can be formed faster [28]. Thus, these alternative ubiquitin conjugations cannot only be used for lysine-less substrates but might also be preferable when a quick response is needed.

Ubiquitination is multifaceted (Figure 1b). The substrate protein can be tagged with just one ubiquitin (monoubiquitination), multiple single ubiquitins (multi-monoubiquitination), or polyubiquitin chains (polyubiquitination). Since ubiquitin itself has seven lysines (Lys6, Lys11, Lys27, Lys29, Lys33, Lys48, and Lys63) and one N-terminal methionine residue (M1) that can be subjected to ubiquitination, several structurally different homogeneous/homotypic, mixed/heterotypic, and branched polyubiquitin chains can be formed [29,30]. Additionally, given that ubiquitin can also be acetylated, phosphorylated, or modified by small ubiquitin-like modifier (SUMO) and neural-precursor-cell-expressed developmentally down-regulated 8 (NEDD8) proteins, multiple modified forms of ubiquitin chains can also be produced [31]. There are currently two main models for how ubiquitin chains are linked to the substrate. The predominant is the sequential model, where individual ubiquitin molecules are transferred and linked to the substrate one molecule at a time [32]. Alternatively, a ubiquitin chain that is pre-assembled on the E2 or E3 enzymes is transferred to the substrate in only one step, as described in the en bloc assembly model [32].

Ubiquitination, aside from being the master regulator of protein degradation via the proteosome, is in fact a multifunctional regulator of nearly every cellular signaling cascade [19]. The fate of the ubiquitinated protein depends on the number of added ubiquitins and the ubiquitin amino acid forming the Ub-Ub linkages [29]. This way, and thanks to the numerous possibilities of topologically distinct chains, ubiquitin constitutes a highly versatile signal. Despite all the homotypic linkage chains being identified in eukaryotic cells, and recently in some heterotypic and branched ones as well, due to the ever-growing complexity and the lack of technical tools to study ubiquitin chains, we are only starting to decode the still enigmatic ubiquitin code (Figure 1b) [30].

Monoubiquitination and multi-monoubiquitination are involved in the endocytosis of membrane receptors, as well as protein degradation, localization, and protein–protein interaction [33,34]. These types of chains have also been implicated in TGF-β signaling [35,36], and their absence has been linked to the pathogenesis of genetic disorders [37]. Among the homotypic polyubiquitin chains, lysine 48 (Lys48)- and lysine 63 (Lys63)-linked chains are the most abundant and studied ones, and they drive proteasomal degradation and regulate diverse cell-signaling events, respectively. Lys11-linked chains regulate cell cycle [38] and mitophagy [39], and linear methionine 1 (M1)-linked chains are essential for inflammatory innate responses, particularly regulating NF-κB signaling [40]. On the contrary, the cellular processes regulated by Lys6-, Lys27-, Lys29-, Lys33-ubiquitin-linkage chains remain less understood. Lys27-linked chains appear important for regulating innate immunity [41], cell cycle [42], and mitochondrial functions [43]. Finally, Lys6-chains have been described to participate in DNA repair [44] and mitophagy [45], whereas, Lys29-chains have been associated with Wnt signaling [46], and just recently, proteotoxic stress and cell cycle [47]. Lys33-chains play a crucial role in post-Golgi trafficking [48] and T-cell receptor (TCR) regulation [49,50]. As for heterotypic chains, Lys11/Lys48-linked chains are found to regulate, in a proteolysis-dependent manner, cell cycle and protein quality control [51]. Furthermore, whereas Lys48/Lys63 branched ubiquitin chain can regulate NF-κB signaling and trigger protein degradation via the proteasome [52,53], Lys29/Lys48 branched ubiquitination participates in autophagy and proteostasis [54] (Figure 1b).

Importantly, the assembly of the ubiquitin to the substrate can be reversed by the action of deubiquitinating enzymes (DUBs), which hydrolyze and remove ubiquitin from the substrate, making ubiquitination a transient modification [55] (Figure 1a). They are also responsible for maintaining the cellular pool of free ubiquitin by recycling ubiquitin chains at the proteasome and processing the newly synthesized ubiquitin precursors into single ubiquitins [56,57]. DUBs are indispensable for the maintenance of the physiological ubiquitin balance [55]. Deregulation of this balance has been linked to disease pathogenesis in humans, in particular cancer, infections, neurodegeneration, and immune disorders [58]. As a consequence, DUBs have started to receive attention as attractive, druggable targets for inhibitor-based therapies [58,59]. The fact that the human genome codes for 2 ubiquitin-activating E1s [60], around 40 E2s [61], over 600 E3s [62], and almost 100 DUBs [63] further support the vital role of ubiquitin system in physiology.

## 3. Key Mechanism of T-Cell Activation and Inhibition and Its Use in Immunotherapy

### 3.1. T-Cell Activation

The first essential signal for T-cell activation is delivered via the TCR-CD3 complex at the cell surface and it is related to antigen-specificity. The TCR recognizes, using CD4 and CD8 co-receptors, an antigen peptide loaded on the major histocompatibility complex (pMHC), presented on the surface of antigen presenting cells (APCs). Since the TCRs lacks an intrinsic signal-transduction domain to transmit the signal, it non-covalently associates itself with the adaptor receptor CD3, forming a functional TCR-CD3 complex [64,65]. The CD3 receptor is composed of CD3εγ and CD3εδ heterodimers, and a CD3ζζ homodimer, which has multiple activating signaling motifs called immunoreceptor tyrosine-based activation motif (ITAM) [66]. The ligation of TCR to antigens leads to phosphorylation-dependent activation of the tyrosine kinase LCK and its recruitment to the TCR via interaction with either CD4 or CD8, depending on the T-cell subtype. From there, LCK phosphorylates the ITAMs in the TCR-CD3 receptor complex, to first recruit the Zeta chain of T-cell receptor-associated protein Kinase 70 (ZAP70) [67]. ZAP70 is then responsible to phosphorylate several scaffolding proteins, particularly LAT and SLP-76, which ultimately drive the phosphorylation-dependent activation of PLCγ1 by the tyrosine kinase ITK [68]. Activated PLCγ1 hydrolyses membrane lipid phosphatidylinositol-4,5-biphosphate (PIP2) to inositol-1,4,5-triphosphate (IP3) and diacylglycerol (DAG) [69]. These secondary messengers amplify the TCR signal primarily via PKCθ, Ras, calcium signaling, and the CARMA1-BCL10-MALT1 (CBM) signalosome complex [70,71,72]. Collectively, these pathways lead to the nuclear translocation of NFAT, AP-1, and NF-κB to drive the genetic programme that guides T-cell proliferation, activation, and survival.

A second signal is needed to activate naïve T cells. This signal is mediated by the interaction between surface co-stimulatory receptors on T cells and their cognate ligands on the APCs [73]. T-cell co-receptors, as well as their ligands, are versatile and are upregulated on the surface of APCs or T cells upon cell activation. This provides context and enables T cells to become activated only when stimulated with non-self or altered self-molecules. The second signal acts to reinforce the signaling pathways first triggered by TCR engagement as well as the immune synapse formed at the contact site between the T cell and the APC, concentrating receptors and signaling molecules in microclusters to facilitate efficient intracellular signaling [74]. This is essential to organize the signaling molecules and TCR receptors temporally and spatially for efficient T-cell activation.

CD28 is the chief co-stimulatory receptor for naïve T cells. It belongs to the Ig superfamily of co-receptors, same as its ligands, CD80 (B7.1) and CD86 (B7.2), which are expressed on activated APCs [75]. CD28 activation signals mainly via phosphoinositide 3-kinase (PI3K)/Akt, as well as the CBM multiprotein complex to induce NF-κB and AP-1 activation, and via the Vav1/Rac1/Cdc42 pathway for controlling the cytoskeleton [76,77]. By amplifying these indispensable TCR signaling events, CD28 lowers the T-cell activation threshold and essentially participates in cytokine production, cell survival, cellular metabolism, cell cycle progression, and cytoskeleton rearrangements to increase plasma fluidity and reinforce a functional immune synapse [78,79]. Other important co-receptors are OX40, 4-1BB, ICOS, GITR, and CD27, which are usually expressed upon T-cell activation to regulate different aspects of T cells, including proliferation, the effector functions, generation of T-cell memory, induction of specific T-cell lineages, and importantly, similar to CD28, can also impact anti-tumor responses [80]. 4-1BB and OX40 are co-stimulatory receptors belonging to TNFR family and are expressed on activated T cells. They play vital role in regulating T-cell proliferation, survival, and cytotoxic functions [81]. Because of this, their use and activation is currently extensively researched for boosting cancer immunotherapies [82], including CAR-T therapies, specifically for 4-1BB and OX40 [82,83]. The inclusion of the signaling domain of 4-1BB in chimeric antigen receptor-T (CAR-T) cells can enhance T-cell persistence and cytotoxic functions, which correlate with reducing tumor burden when adoptively transferred to tumor-bearing mice [84]. The transplantation, into mouse tumor models, of CAR-T cells that in addition to 4-1BB constitutively expressed OX40 as a co-receptor further enhanced T-cell proliferation and survival, anti-tumor cytotoxicity, and better prevented their exhaustion; similar benefits were observed when transplanted into metastatic lymphoma patients [83].

### 3.2. T-Cell Inhibition

Immediately after TCR activation, several inhibitory mechanisms are triggered. This negative feedback is essential for immune balance, limiting the duration and amplitude of the T-cell response to guarantee tissue repair and homeostasis. Co-inhibitory receptors are core mechanisms of T-cell attenuation; hence, they are known as immune checkpoints. One of the most prominent co-inhibitory checkpoint receptor is the cytotoxic T-lymphocyte-associated protein 4, CTLA-4, a CD28 homolog that has a higher affinity toward CD80 and CD86 ligands in the activated APCs [85]. Very soon after TCR engagement, CTLA-4 is expressed as well as translocated to the cell surface from its intracellular vesicle reservoir [86]. Once there, it outcompetes CD28 for its ligands; this competitive hindrance on CD28 signaling is believed to be its major mechanisms of immune suppression [87]. CTLA-4 is also able to down-regulate the availability of CD28 ligands by mediating the trans-endocytosis of CD80 and CD86 [88] and can further recruit phosphatase SHP1 to shutdown phosphorylation events downstream of the TCR/CD28 [89].

The other master co-inhibitory receptor is the programmed cell death protein 1, PD-1, which is also a CD28 family member and binds to two ligands PD-L1 (B7-H1, CD274) as well as PD-L2 (B7-DC, CD273), expressed at the surface of other immune cells as well as non-lymphoid tissues, including tumor cells [90]. Upon binding, the PD-1/PD-L1 inhibitory pair functions to limit T-cells responses and lessen the tissue damage. Mechanistically, PD-1 engagement leads to phosphorylation of its immune receptor tyrosine-based inhibitory (ITIM) and immune receptor tyrosine-based switch (ITSM) intracellular motifs [91]. The SHP2 phosphatase is then recruited to these phosphorylation sites from where it can dephosphorylate key mediators of TCR proximal signaling, such as ZAP70, and as well as downstream of CD28, in particular PI3K [92].

There are other functionally important co-inhibitory receptors in T cells. For instance, TIM3, which promotes immunosuppression by positively regulating the suppressive functions of Tregs and the expansion of myeloid suppressor cells [93,94]. TIM3 blockade in mice leads to spontaneous autoimmune disorders [95]. The co-inhibitory receptor TIGIT is also expressed upon T-cell activation to bind CD112, CD113, and CD155 ligands on APC and cancer cells; TIGIT inhibition can improve anti-tumor immune responses in vivo [96].

The stability of the receptors and the proximal signalosome at the cell surface is another regulatory layer in T-cell activation. Upon T-cell activation, inhibitory molecules start to shut down TCR signaling, including the dephosphorylation of LCK by the CD45 membrane-bound phosphatase [97]. In addition, the activated TCR-CD3 complex, as well as CD28 co-receptors, are internalized from the immunological synapse via clathrin-dependent and/or independent pathways for degradation, ultimately leading to T-cell down-regulation [98]. During their internalization, key proximal signalosome components such as SLP76 and ZAP70 are segregated from the receptors and targeted for degradation [99]. Bystander TCR-CD3 complexes are also internalized to be recycled and recruited back to the immunological synapse [100,101].

T-cell stimulation in non-optimal conditions also prevents full sustainable T-cell activation, as an additional mechanism of immunosuppression. Incomplete T-cell stimulation in the absence of a co-stimulatory signal triggers a suppressive genetic program that forces T cells to acquire a long-term state of unresponsiveness called anergy [102]. This is a core mechanism of peripheral tolerance. On the other extreme, excessive or constant activation of T cells in the context of chronic viral infections or cancer often triggers T-cell dysfunction, where T cells end up instead hypo-responsive [103,104]. Whereas constant antigen stimulation usually leads to T-cell exhaustion [104], T cells experiencing repeated antigen-driven proliferation can enter a state of senescence and become cell cycle arrested [105]. Whereas exhausted T cells typically co-express several inhibitory receptors such as CTLA-4, PD-1, LAG-3, and TIM3, senescent T cells often down-regulate co-stimulatory molecules, including CD28 and CD27 [106,107,108]. Anergized, exhausted, and senescent tumor-infiltrating immune cells are commonly found in the tumor microenvironment of mice and cancer patients, and are key obstacles for the efficacy of cancer immunotherapy [109,110].

### 3.3. T-Cell Targeted Cancer Immunotherapy

Current T-cell targeted cancer immunotherapy protocols leverage the most proximal events at the cell surface of T cells to enhance and reconstitute tumoral immune responses. This is accomplished either by blocking the receptor-ligand interaction of key inhibitory surface co-receptors, such as CTLA-4 and PD-1, a therapy known as checkpoint inhibition [111], or potentiating T-cell activation by using gene-edited and improved T-cell receptors, especially chimeric antigen receptor (CAR) T-cell therapy [112]. Currently, there are eight FDA-approved immune checkpoint therapies (Ipilimumab- targeting CTLA-4, Nivolumab, Pembrolizumab, and Cemiplimab targeting PD-1, Atezolizumab, Avelumab, Durvalumab targeting PD-L1 and the combinational therapy of Ipilimumab + Nivolumab) as well as five CAR-T immunotherapies (Abecma^®^ targets BCMA, and Breyanzi^®^, Tecartus^TM^, Kymriah^TM^, Yescarta^TM^ targeting CD19). Whereas immune checkpoint inhibitors are used for treating multiple solid tumors, CAR-T therapies, so far, are used for hematological malignancies.

The checkpoint strategy utilizes blocking antibodies to target the inhibitory receptor or the ligands, thus, preventing their interaction. By blocking CTLA-4, CD28 is guaranteed to prolong access to their B7.1 and B7.2 ligands during T-cell priming [87]. Given the cell-extrinsic suppressive function of CTLA-4 in Tregs, CTLA-4 targeting therapies have additional immunostimulatory effects by acting on Tregs [87]. Since PD-L1 ligands are expressed on tumor cells in an inflammatory environment and as an important mechanism of immunoescape, PD-1/PD-L1 blockade therapy primarily act at the tumor site to overcome tumor-induced suppression of immune cells. Specifically, PD-1/PD-L1 blocking antibodies lift TCR from obstructive PD-1 pathways, relieving T cells from exhaustion and reinvigorating T-cell responses at the tumor site [113,114,115].

On the other hand, CAR-T therapies use a distinct and novel approach to exploit TCR signaling for tumor rejection [112]. The patient’s T cells are isolated and engineered, before being transferred back, to express a chimeric antigen receptor (CAR). Extracellularly, the CAR carries an antibody fragment that recognizes specific antigens directly at the surface of tumor cells, making T-cell activation via CARs now independent of MHC presentation. The intracellular signaling domain combines the CD3ζζ ITAMs of the TCR/CD3 complex (signal 1) with the intracellular domains of co-stimulatory receptors (signal 2, either CD28 or 4-1BB). Researchers working on the next generations of CARs, are exploring the benefits of dual expression of co-stimulatory receptors (CD28-4-1BB or CD28-OX40), constitutive or inducible expression of key T-cell interleukins (IL-2, IL-15, IL-12), as well as suicidal genes or molecular switches for safer on-off manipulation of T cells [116].

Despite their clinical benefits, T-cell-based immunotherapies face significant hurdles [117,118]. They fail to benefit most patients, and still encounter difficulties enhancing and sustaining the activation, proliferation, infiltration, and long-term survival of T cells. Further, tumors often develop new compensatory mechanisms to evade these targeted therapies and relapse. Multiple clinical and preclinical efforts are underway to address these significant clinical challenges [119,120]. Checkpoint and CAR-T therapies work at the receptor level; thus, targeting the downstream signaling cascades and/or the stability of those co-inhibitors and CAR receptors can offer interesting alternative protocols. Ubiquitin-related enzymes are master regulators of protein homeostasis and signaling cascades in T cells and could therefore provide the molecular basis for such effective forthcoming therapies.

## 4. Ubiquitination Is Essential to Regulate T Cell Activating and Inhibitory Signaling

### 4.1. Ubiquitin-Pathways in TCR/CD3 Activation

#### 4.1.1. E3 Ligases

The E3 ubiquitin ligase Casitas B-lineage lymphoma (Cbl-b) is the best-characterized E3 ligase as a principal gatekeeper of T-cell activation. It is recruited to the TCR at the immunological synapse upon T-cell activation [121], where it exerts numerous inhibitory mechanisms downstream the TCR. Cbl-b interacts, via its multiple protein-interacting domains, with key TCR signalosome molecules such as LCK, SLP76, ZAP70, Vav1, PKCθ, and PI3K [122]. Cbl-b, in coordination with the E3 ligases Itch, is able to mediate the Lys33-linked poly-ubiquitylation of TCR-ζ (Figure 2), which does not target the TCR receptor for degradation or endocytosis but, instead, prevents its phosphorylation and further association with the downstream ZAP70 kinase [49]. Through these interactions, and ubiquitination of some of these signalosome components, so far reported for the p85 regulatory subunit of PI3K and PLCγ1 (Figure 2), but presumably more, Cbl-b strongly dampens T-cell activation [123]. Consequently, mice deficient in Cbl-b have hyperactive T cells that do not require CD28 for their activation [124], which significantly enhances T-cell anti-tumor immunity in vivo [125]. In fact, the absence of Cbl-b allows mice to consistently and spontaneously reject dozens of diverse tumors, including transplantable, UV-induced, metastatic, and genetically driven tumors [125,126,127,128,129], and display long-term tumor-specific immunological memory [126]. *Cbl-b*^−/−^ CD8^+^ T cells can also confer anti-tumor activity and reject tumors if adoptively transferred to tumor-bearing mice [125,126,130]. *Cbl-b*^−/−^ mice have less exhausted T lymphocytes present in the tumor environment, and targeted depletion of Cbl-b, via CRISPR/Cas9, can restore the expression of inflammatory cytokines and cytotoxic molecules in wild-type exhausted PD1^+^Tim3^+^ T cells [131]. Importantly, absence of Cbl-b could also render T cells resistant to the tolerogenic tumor microenvironment, since T cells from *Cbl-b*^−/−^ mice cannot be anergized, neither in vitro upon incubation with ionomycin nor in vivo, in P14 or OTII TCR transgenic mouse models of anergy [132,133], and are resistant to Treg and TGF-β suppression [125,126,134] well as to PD-1 or CTLA-4 inhibition [135,136]. Although *Cbl-b* knockout mice have higher susceptibility to diverse experimental models of autoimmunity, including encephalomyelitis [124], autoimmune arthritis [133], and type 1 diabetes [137], spontaneously, Cbl-b-deficient mice, including knockout and catalytic dead *knockin* mice, only develop a mild and non-lethal autoimmunity phenotype [126,138]. Importantly, no signs of autoimmune toxicity have been reported in Cbl-b-deficient mice challenged with tumors, neither short-term, while rejecting tumors, nor long-term, up to 1 year after tumor rejection [125,126,127,129,138]. Similarly, no autoimmune injury was ever described in wild-type mice receiving *Cbl-b* knockout or knockdown CD8^+^ T-cell-based adoptive transfer immunotherapy [125,126,128,130,139,140], not even when the same tumor antigens are expressed in distal organs [128]. Thus, for its multiple checkpoint inhibitory roles and minimal autoimmune toxicities, Cbl-b is a strong candidate for future targeted cancer immunotherapies.

The gene related to anergy in lymphocytes (GRAIL), is a transmembrane E3 ligase localized in endosomes that also negatively regulates T-cell activation. TCR-mediated activation of T cells promotes the expression of GRAIL, which in turn hinders T-cell activation [141]. While overexpression of GRAIL reduces IL-2 and IL-4 secretion by T cells [142], T cells that lack GRAIL display both enhanced proliferation and secretion of cytokines independent of CD28 co-stimulation [141,143]. Mechanistically, GRAIL stabilizes and activates, via non-degradative ubiquitin chains, the Rho guanine dissociation inhibitor (RhoGDI) (Figure 2). This is hypothesized to impair the Rho signaling pathways leading to defects in cytoskeleton rearrangement and IL-2 secretion [144]. NRDP1 is another E3 ligase that impairs TCR signaling. Upon T-cell activation, NRDP1 adds Lys33-polyubiquitin non-degradative chains to the ZAP70 kinase (Figure 2). Lys33-linked ubiquitin chains help recruit Sts1 and Sts2 phosphatases, which ultimately dephosphorylate and inactivate ZAP70 [50]. Similar to Cbl-b and GRAIL-deficient T cells, CD8^+^ T cells isolated from *Nrdp1*^−/−^ mice show increased capacity to proliferate and produce IL-2 and IFN-γ [50]. These NRDP1 inhibitory T-cell effects are counteracted by the deubiquitinase OTUD7B. By deubiquitinating ZAP70 (Figure 2), OTUD7B maintains ZAP70 in a phosphorylated and active state [145]. Similarly, the ubiquitin specific peptidase 12 (USP12) can deubiquitinate LAT at the proximal TCR (Figure 2), to protect it from ubiquitin-dependent lysosomal degradation [146]. Thus, USP12 and OTUD7B counterbalance inhibitory E3 ligases to stabilize TCR signaling.

Aside from acting at the proximal signalosome, E3 ligases also inhibit T-cell activation by acting further downstream, controlling the key transduction events that lead to the nuclear translocation of the T-cell activating transcription factors. For example, upon TCR/CD28 ligation, the E3 ligase Pellino 1 (PELI1) prevents NF-κB pathway by tagging c-Rel with Lys48-polyubiquitin chains, leading to its proteasomal degradation (Figure 2). As a consequence, *Peli1*^−/−^ mice have hyper-responsive effector T cells [147]. The HECT E3 ubiquitin ligases ITCH and NEDD4 also participate in attenuating NF-κB-mediated T-cell activation. For this, ITCH and NEDD4 ubiquitinate BCL10 from the CARMA1-BCL10-MALT1 (CBM) complex resulting in its lysosomal degradation [148] (Figure 2). The E3 ligase MDM2 inhibits CD4^+^ T-cell activation by preventing NFATc2 activation and cytokine production in a p53-independent manner [149]. This negative regulatory mechanism is reinforced by the activity of the deubiquitinating enzyme USP15, which deubiquitinates degradative Lys48-chains from MDM2 to stabilize it (Figure 2). USP15-deficient mice have T cells with elevated expression of CD44, an effector T-cell marker, and can secrete higher levels of IL-2 and IFN-γ. Because of this, deficiency of USP15 serves to promote anti-tumor response in T cells [149].

#### 4.1.2. DUBs

The deubiquitinases A20 and CYLD, mostly investigated for their essential role in the regulation of NF-κB signaling in innate immune cells [150], can also significantly alter T-cell function. Conditional deletion of A20 in peripheral T cells leads to CD8^+^ T cells with augmented NF-κB signaling that could become activated more efficiently, even with low antigen doses, to produce higher levels of IL-2 and IFN-γ [150,151]. Adoptive transfer of A20-deficient cytotoxic T cells to tumor-bearing mice resulted in smaller tumors, with more infiltrating T cells that expressed levels of PD-1 [151]. A20 was reported to inhibit NF-κB signaling by deubiquitylating the CBM complex molecule MALT1, and thereby inhibiting the interaction of MALT1 with the IKK complex required for NF-κB signaling [152] (Figure 2). The deubiquitinase CYLD also impairs T-cell activation via NF-κB, but in this case by deubiquitinating Lys63-ubiquitin chains from the TAK1 kinase downstream of the CBM complex (Figure 2). CYLD-deficient mice present with colitis and increased T-cell frequency and activation [153]. Deficiency of CYLD leads to hyper Lys63-ubiquitination of TAK1 and, therefore, TAK1 hyperactivation with associated spontaneous activation of IKK, and eventually, NF-κB [153]. Using similar mechanisms of action, deubiquitination of TAK1 by the deubiquitinase USP18 inhibits TCR (Figure 2). Genetic depletion of USP18 causes NF-κB and NFAT hyperactivation and hyperproduction of IL-2 in T cells [154]. On the contrary, the deubiquitinases USP9X and USP12 positively regulate NF-κB pathways through removal of inhibitory ubiquitin chains from BCL10, which block the assembly of the CBM signaling complex (Figure 2). Consequently, T cells deficient of USP9X or USP12 display lower levels of NF-κB activation as well as reduced proliferation and cytokine production upon TCR activation [155,156].

#### 4.1.3. TCR Internalization

TCR signaling amplitude and duration is determined by the balance of internalization, recycling, and degradation of receptors from the cell membrane. Although it has long been known that upon antigen-stimulation TCR/CD3 complexes are ubiquitinated [157,158] and that ubiquitination is essential for the internalization of surface receptors [159] we are only starting to reveal the ubiquitin-dependent mechanisms controlling TCR endocytosis and stability at the cell surface. Cbl-b is reported to promote TCR internalization, either alone or in combination with the E3 ligase c-Cbl [160]. Cbl-b additionally interferes with activation and clustering, thus destabilizing the immune synapse, further attenuating TCR signaling [161]. Likewise, the E3 ligase GRAIL can down-regulate the expression of the TCR/CD3 complex at the surface by poly-ubiquitinating and degrading CD3ζ molecules via the proteasome [141] (Figure 2). It was recently shown that the surface expression of CAR receptors used in CAR-T therapy is also affected by ubiquitin-mediated endocytosis and degradation [162]. Poor persistence of infused CAR-T cells into the patient is the major limitation of this cancer immunotherapy [163]. A recent study was able to improve CAR stability by mutation at all lysine residues of the cytoplasmic domain [162]. This efficiently bypasses the ubiquitination-dependent lysosomal degradation of the CARs, which could be readily recycled back to the surface. As a result, these CAR-T cells have improved CAR persistence and signaling from the endosomes, coupled with enhanced effector functions leading to a long-lasting anti-cancer immune response in mouse models [162].

### 4.2. Ubiquitin-Pathways Downstream of Co-Stimulatory Receptors

#### 4.2.1. CD28

The E3 ligase Cbl-b is a major inhibitor of CD28-mediated signaling. This is functionally supported by the fact that T cells lacking Cbl-b become activated without CD28 [122] and Cbl-b deficiency can rescue, in vivo, most of the T-cell alterations observed in CD28 knockout mice [124]. Mechanistically, the E3 ligase Cbl-b inhibits CD28-dependent PI3K pathway by binding and ubiquitinating the p85β subunit of PI3K [164] (Figure 2). The ubiquitination chains are non-degradative in nature, but inhibit p85β from interacting with CD28 [165]. *Cbl-b*^−/−^ T cells present increased Akt activation and NF-κB signaling, which was once thought to be a result of the inhibitory effect of Cbl-b on p85 [166]. However, a later study revealed that Cbl-b inhibits PI3K indirectly by preventing the TCR/CD28 cascades from inactivating PTEN phosphatase via NEDD4-dependent ubiquitination. Co-triggering of TCR and CD28 induces the E3 ligase NEDD4 to bind and inactivate PTEN with Lys63-ubiquitin chains; Cbl-b acting on NEDD4 impairs its association and ligase activity toward PTEN [167] (Figure 2). Collectively, the multi-level regulatory role of Cbl-b on PI3K prevents the activation of Vav1/Cdc42 pathways [161,168] which are essential for efficient cytoskeleton rearrangement, as well as activation of the CBM signaling complex required for NF-κB activation [166].

E3 ligases can not only hinder CD28 signaling but also promote it. For instance, activation of NFAT by TCR/CD28 involves the E3 ligase TRAF6. TRAF6 is recruited to the immunological synapse to interact with the scaffolding protein LAT and, by Lys63-ubiquitination (Figure 2), enhance LAT phosphorylation and TCR signaling [169]. For T cells to enter the replicative S phase of the cell cycle, CD28 receptors induce the expression of SKP2, the substrate recognition component of the SCF E3 ligase complex. SCF^Skp2^ then targets the cyclin-dependent kinase inhibitor 1B (KIP1) with degradative ubiquitin chains (Figure 2). Thus, CD28 ubiquitin-dependent down-regulation of KIP1 permits cell cycle regulation in T cells [170].

To fully activate T cells, CD28 co-receptors must overcome Cbl-b inhibition. CD28 signaling disables Cbl-b inhibitory pathways by triggering post-translational modifications on Cbl-b that ultimately lead to its proteasomal degradation [136,171]. In order to degrade Cbl-b, CD28 promotes its phosphorylation in two ways; it prevents the SHP1 phosphatase from dephosphorylating Cbl-b, a mechanism that was initially triggered by the TCR/CD3 [172], and further induces the phosphorylation of Cbl-b by LCK [172] and the PKCθ kinase [173] (Figure 2). It was hypothesized that Cbl-b phosphorylation promotes a conformational change that, in turn, may result in Cbl-b ubiquitination and proteasomal degradation [173]. Although not yet directly linked, the E3 ligase NEDD4 is a potential candidate to facilitate Cbl-b ubiquitination upon these phosphorylation events, given that NEDD4 has been reported to bind and ubiquitinylate Cbl-b for proteasomal degradation upon CD28 co-stimulation [174] (Figure 2). CD28-induced down-regulation of Cbl-b does not occur in the absence of p85β [164] or in aged mice [175], possibly due to age-dependent defects in the proteasome system. Notably, not all phosphorylation events on Cbl-b via PI3K pathways lead to its degradation. The GSK3 kinase phosphorylates Cbl-b at the S476 and S480 sites to stabilize it. In turn, PI3K, via Akt, is reported to down-regulate Cbl-b levels by inhibiting the activity of GSK3 [176]. TCR/CD28 signaling also bypass the inhibitory pathways of the E3 ligase GRAIL [177]. For this, CD28 co-stimulation via IL-2/mTOR-dependent pathways induces the expression of the deubiquitinase Otubain-1 (OTUB1) [177], which destabilize and promote GRAIL degradation [178] (Figure 2). Additionally, CD28-costimulation overcomes the NF-κB-inhibitory deubiquitinase A20 at the CBM signaling complex, by inducing A20 cleavage by the paracaspase MALT1 [179].

#### 4.2.2. Other Co-Stimulatory Receptors

Similar to CD28, the co-stimulatory receptors 4-1BB, CD40L, OX40, and GITR, members of the tumor necrosis factor receptor (TNFR) family, synergize with the TCR to promote T-cell activation, particularly proliferation and cytokine production [81]. As TNFRs, they lack intrinsic enzymatic function and depend on several E3 ligases, including TRAFs (TRAF1/2/3), cIAP1/2, and LUBAC, to guide numerous Lys63, Lys48, and linear ubiquitination events along the signal transduction cascade leading to NF-κB activation. The deubiquitinases A20, CYLD, and OTULIN counteract these ubiquitination events to impair NF-κB activation [81,180,181,182] (Figure 2).

Compared to CD28, CARs using 4-1BB for co-stimulation are reported to better promote metabolic pathways and recovery from exhaustion [183,184]. Additional to requiring TRAFs E3 ligases for assembly of its proximal signalosome, the recruitment of 4-1BB to the lipid rafts can trigger important signaling events, presumably via Akt, after its endocytosis [180]. Lys63-type ubiquitination by the E3 ligase TRAF2 is required for 4-1BB receptor internalization, its subsequent signaling from endosomes, and ultimately, for 4-1BB-dependent tumor rejection [185]. When treated with agonist anti-4-1BB immunotherapy, TRAF2-deficient mice displayed delayed and less efficient tumor rejection, consequently carrying larger subcutaneous CT26 colorectal tumors [185]. The deubiquitinases A20 and CYLD interact with the 4-1BB and TRAF2 complex and, by regulating the ubiquitination of TRAF2 and TAK1 (Figure 2), they decrease 4-1BB activation [186]. In primary human cytotoxic CD8^+^ T cells, the knockdown of A20 or CYLD augmented the effects of 4-1BB co-stimulation, leading to higher protein levels of CD25 and the anti-apoptotic Bcl-xL molecule [186].

Signaling via the co-stimulatory receptor GITR is important for T-cell anti-tumor responses. In mice, stimulation of this receptor with anti-GITR agonists was shown to reverse T-cell exhaustion and deplete Tregs, leading to strong anti-tumor activities [187]. A recent study revealed that the expression of the co-stimulatory receptor GITR is regulated by the E3 ligase NEDD4 (Figure 2). NEDD4 ubiquitin-mediated proteasomal degradation of GITR diminishes, in vitro, the cytotoxic response of Jurkat T cells toward melanoma cells [188]. The expression of ICOS, CD40L, and OX40, three co-stimulatory receptors essential for T follicular helper cell (Tfh) differentiation and function [189] (Figure 2), and of relevance in cancer immunotherapy [190,191], is regulated by E3 ligases. The E3 ligase PELI1 negatively regulates Tfh differentiation and function by ubiquitinating the transcription factor c-Rel for degradation (Figure 2), which then fails to induce the expression of ICOS and CD40L [192,193]. During anergy induction, the up-regulation of the transmembrane E3 ligase GRAIL correlated with decreased expression of CD40L, since GRAIL binds to CD40L and ubiquitinates it for degradation [194] (Figure 2). The paralogs Roquin-1 and Roquin-2, interesting mRNA binding proteins carrying an active RING-E3 ligase domain, also decrease ICOS expression by acting directly at the mRNA levels (Figure 2). Roquin directly binds to the 3′ untranslated region (3′UTR) of ICOS mRNA [195,196], assisted by the cofactor NUFIP2 [197]. Likewise, both Roquin paralogs down-regulate OX40 levels by repressing OX40 mRNA levels [198] (Figure 2). T cells from Roquin-deficient mice show elevated levels of ICOS and OX40. Moreover, these mice exhibit an excessive number of Tfhs and germinal centers [198,199]. Interestingly, the elevated levels of ICOS could override the requirement for CD28-costimulation in the T cells of Roquin-deficient mice [199].

### 4.3. Ubiquitin-Pathways Downstream CTLA-4 and PD-1/PD-L1 Co-Inhibitory Receptors

Up to now, direct ubiquitination of CTLA-4 by any E3 has not been reported, and there are no records of ubiquitinated sites in CTLA-4. Nevertheless, CTLA-4-deficient T cells present a significant decrease in their overall ubiquitin modifications [200], strongly indicating the importance of ubiquitination events in CTLA-4 pathways. Moreover, experimental evidence indicates that major CTLA-4 inhibitory functions are mediated by the key T-cell inhibitory E3 ligases that also regulate T-cell activation, namely Cbl-b, ITCH, and GRAIL (Figure 3). CTLA-4-deficient T cells have reduced levels of Cbl-b, while CTLA-4 activation up-regulates Cbl-b mRNA levels [136] (Figure 3). CTLA-4 positively regulates the activity of ITCH post-translationally, by inducing its dephosphorylation [200]. Activation of CTLA-4 receptors increases GRAIL levels by repressing the expression of the deubiquitinase Otubain-1 [177] (Figure 3), which controls GRAIL degradation [178]. Functionally, the inhibitory effects of CTLA-4 are relieved by depletion of either Cbl-b or ITCH [136,200]. Whether these E3 ligases function to inhibit TCR/CD28-driven T-cell activation via previously described pathways, or alternatively, drive novel signaling cascades downstream of CTLA-4 needs to be determined. Additionally, Cbl-b, ITCH, and GRAIL are the essential drivers of T-cell anergy; in their absence, T cells cannot be anergized in vitro nor in vivo [201]. Further, these three E3 ligases also play essential immunosuppressing roles by regulating Tregs development and functions [201,202], where CTLA-4 is also known to have critical functional effects, particularly important for cancer immunotherapy [203]. Cbl-b and ITCH participate in TGF-β mediated regulation of Foxp3, the absence of either of these E3 ligases impairs the development of TGF-β induced Foxp3^+^ Tregs (iTreg), resulting in iTregs that have low Foxp3 expression and are functionally defective, unable to suppress T-cell proliferation and airway inflammation, respectively [204,205,206,207]. GRAIL overexpression can convert T cells to a regulatory phenotype with cellular markers of Treg and immusupressive activities [208,209]. Although yet to be experimentally tested, it is possible that CTLA-4 participates in anergy and Treg functions utilizing these same E3 ligases. Future and detailed molecular studies on CTLA-4 ubiquitin-dependent pathways could help leverage ubiquitination-targeted therapy as an alternative or complement to CTLA-4 immunotherapy.

Unlike CTLA-4, several reports highlight the critical role of ubiquitination in the PD-1/PD-L1 system (Figure 3). For the PD-1 receptors on the T cells, Cbl-b, not surprisingly, also appears significantly involved as a key mediator of the PD-1 inhibitory pathway. T cells lacking Cbl-b are resistant to PD-1 inhibition [135]. Without Cbl-b, PD-1 is inefficient in suppressing IFN-γ production or inducing cell death upon T-cell activation, and *Cbl-b*^−/−^ mice are capable of rejecting melanoma tumors that escape immune response in wild-type mice via PD-1 signaling [135]. Furthermore, PD-1 engagement is required to induce the TCR-dependent up-regulation of Cbl-b protein levels (Figure 3), which associates with a concomitant internalization of TCR surface receptors [210]. In mice, Cbl-b was found to be up-regulated in the PD-1^+^Tim3^+^ exhausted CD8^+^ T cells infiltrating MC38 colon tumor, and *Cbl-b*^−/−^ mice had reduced numbers of exhausted T cells in the tumor microenvironment. Importantly, CRISPR/Cas9-mediated depletion of *Cbl-b* in sort-isolated PD-1^+^Tim3^+^ T cells could largely restore CD8^+^ effector functions and absence of Cbl-b (*Cbl-b*^−/−^) in CAR-T cells specific for the human carcinoembryonic antigen (hCEA) increased overall mice survival, enhanced anti-tumor activity, and resulted in less PD1^+^Tim3^+^-exhausted tumor infiltrating CAR-T cells when transferred to mice carrying hCEA-expressing MC38 tumors [131]. The other Cbl E3 ligase, c-Cbl, also affects PD-1, but contrary to Cbl-b, as a negative regulator. c-Cbl binds to the cytosolic tail of PD-1 and, acting as an E3 ligase, ubiquitinates PD-1 for its proteasomal degradation [211] (Figure 3). Consequently, genetic reduction of c-Cbl (*c-Cbl*^+/−^) elevates PD-1 expression in CD8^+^ T cells and macrophages [211]. Likewise, the E3 ligase FBXO38 controls T-cell anti-tumor responses by mediating PD-1 degradation. Mechanistically, FBXO38 tags PD-1 for degradation by adding Lys48-polyubiquitin chains at the Lys233 [212] (Figure 3). Melanoma and colorectal cancer mice models deficient in FBXO38 have higher tumor burden which related with higher PD-1 expression in their tumor-infiltrating T cells; PD-1 blockade rescued anti-tumor activities [212]. Interestingly, this study provided an additional molecular explanation for the anti-cancer effects of interleukin 2 (IL-2) therapy. Administration of IL-2 into wild-type mice enhanced FBXO38 levels in tumors-infiltrating T cells and down-regulated PD-1 from their surface, resulting in more efficient anti-tumor responses, even when IL-2 simultaneously increased, by activating T cells, PD-1 mRNA transcription [212].

The tumor levels of PD-L1 are important determinants of tumor immunity. Several studies have identified that cancer cells use EGFR signaling to stabilize PD-L1 expression for escaping T-cell immunity. The EGFR’s ability to promote PD-L1 expression was shown almost a decade ago in mouse and human studies, where EGFR activation, overexpression or oncogenic mutations correlated with higher PD-L1 levels [213,214,215]. EGFR kinase inhibitors could block the EGF-induced PD-L1 overexpression [216]. Mechanistically, EGF receptors increase PD-L1 through the activation of PI3K/AKT and JAK/STAT3 pathways [216,217] (Figure 3). Interestingly, it was determined that tumors in microenvironments with high levels of copper have elevated PD-L1 expression and that copper can protect PD-L1 from ubiquitin-dependent degradation by promoting the EGFR/STAT3 pathway (Figure 3). On the contrary, copper chelators can inhibit STAT3 and EGFR phosphorylation in cancer cells, to enhance PD-L1 degradation in vitro and in vivo, leading to strong anti-tumor activities [218]. In addition, they also counteract the action of E3 ubiquitin ligases controlling PD-L1 degradation. For E3 ligases to recognize and ubiquitin-tag PD-L1 for proteolysis, PD-L1 must first be phosphorylated, mostly dependent on the glycogen synthase kinase 3 (GSK3α/β). On one side, GSK3β phosphorylates the C-terminal domain of PD-L1 to recruit the E3 ligase β-TrCP for PD-L1 ubiquitin-dependent degradation [219] (Figure 3). The receptor tyrosine kinase MET also participates in this process. MET promotes PD-L1 degradation by phosphorylating GSK3β at its Y56 residue, which prevents GSK3β ubiquitination by TRAF6, leading to higher GSK3β kinase activity [220]. On the contrary, EGFR signaling counteracts the GSK3β/β-TrCP/PD-L1 degradation pathway by inducing substantial glycosylation of PD-L1 next to its phosphorylation site for GSK3β, thus antagonizing PD-L1-GSK3β binding [219] (Figure 3). GSK3β inhibition in vivo resulted in a higher tumor burden which was dependent on PD-1 inhibitory pathways, and PD-1 blocking antibodies restored anti-tumor responses [219].

A study this year further implicated the GSK3α/ARIH1 as another kinase/E3 ligase pair inducing PD-L1 proteolysis. In this case, GSK3α-dependent phosphorylation of PD-L1 at S279/S283 is recognized by the E3 ligase ARIH1, followed by PD-L1 Lys48-ubiquitination and degradation [221] (Figure 3). Whereas immunocompetent mice are able to reject cancer cell lines overexpressing the ARIH1 E3 ligase, immunocompromised mice are not, reinforcing the crucial role of ARIH1-ubiquitin-dependent pathways in anti-tumor immunity [221]. Recently, it was revealed that EGFR increases PD-L1 levels by additionally interfering with the membrane-bound MARCH8 E3 ligase, which also targets PD-L1 for degradation [222] (Figure 3). Finally, c-Cbl overexpression, alone or in combination with Cbl-b, can also decrease STAT3/AKT/ERK phosphorylation to down-regulate PD-L1 expression and increase anti-tumor responses [223]. Whether this is dependent on direct actions of c-Cbl on PD-L1 or the capacity of c-Cbl/Cbl-b to internalize and down-regulate EGFR receptors has not been tested [224,225].

The Cullin3-SPOP and TRIM21 E3 ligases have been shown to further destabilize PD-L1 by ubiquitination-dependent degradation, and their actions are again dependent on kinases, specifically, cyclin-dependent kinases CDK4 and CDK5 [226,227]. Whereas, CDK4 assist in PD-L1 degradation by stabilizing Cullin3-SPOP [226], CDK5 negatively interferes with TRIM21 actions on PD-L1 [227] (Figure 3). Notably, PD-L1 can be protected from ubiquitination not just by glycosylation, but also other post-translational modifications or interactions. PD-L1 palmitoylation, as well as PD-L1 interactions with transmembrane proteins CMTM4 and CMTM6 at the cell surface and endosome membranes, can block E3 ligases, in particular STUB1, from access to PD-L1 ubiquitination sites [228,229,230] (Figure 3). Whether these pathways are connected to EGFR/GSK3 signaling needs to be addressed.

Several deubiquitinases act in opposition to E3 ligases’s PD-L1 degradative functions to maintain PD-L1 levels at the cell surface. The pro-inflammatory cytokine TNF-α induces the expression of the deubiquitinase CSN5 which deubiquitinates PD-L1 for stabilization (Figure 3), revealing a novel mechanism of immune escape under chronic inflammatory microenvironment [231]. The small molecule berberine (BBR), which selectively binds and inhibits CSN5, can diminish PD-L1 expression in cancer cells by enhancing PD-L1 ubiquitin-dependent proteasomal degradation. Berberine also displayed immunoactivating effects in the tumor microenvironment, increasing the frequency of tumor-infiltrating T cells as well as dampening myeloid-derived suppressor cells and Tregs [232]. It was later revealed that another deubiquitinase, USP22, could contribute to CSN5-mediated regulation of PD-L1. USP22 can deubiquitinate PD-L1 itself or CSN5 for their stabilization [233] (Figure 3). PD-L1 is also stabilized owing to the enzymatic activities of several other deubiquitinases, namely OTUB1, USP9X, USP7 [234,235,236] (Figure 3). These studies have potential therapeutic implications as they demonstrate that inhibition, ablation, or knockdown of any of these DUBs can reinvigorate anti-tumor responses in mice and sensitize cancer cells toward T-cell cytotoxicity [234,235,236].

Collectively, the numerous studies on ubiquitin-dependent regulation of TCR/CD3, co-stimulating and co-inhibitory receptors signaling have not only conclusively demonstrated the physiological relevance of these ubiquitin pathways but showed that selectively interfering with key components of these cascades can consistently lead to robust anti-tumor activities in vivo.

## 5. Exploiting Ubiquitin-Dependent Pathways for Cancer Treatment

The first successful clinical trials, now some FDA-approved treatments, targeted ubiquitin-dependent pathways for cancer therapy using small molecule inhibitors to block the proteasome. The characterization of the specific roles of ubiquitin-related enzymes in physiology and disease pathogenesis, coupled with the fact that these enzymes are substrate selective and have great diversity, propelled researchers in the academia and industry to shift interest to selectively target E1s, E2s, E3s, and DUBs. Over 200 compounds have been developed with this purpose and many are currently being tested in preclinical and clinical trials for cancer treatment (Table 1). So far, the predominant technology is based on small molecules that can either block the ubiquitin-related enzyme’s catalytic domain by direct binding or allosterism, antagonize the enzyme, or prevent its binding to substrates or other regulatory proteins.

However, the design or screen for these selective compounds is often challenging due to the structural and functional redundancy among enzymes of the same family, the fact that relevant substrates remain largely unknown and the vast majority of ubiquitin ligases are RING-type E3s (≈600) that do not carry a conventional enzymatic domain with an active catalytic cysteine [21,237]. Additionally, these chemical compounds often require high doses to achieve effective inhibition, leading to off-target effects and cellular toxicities. Novel and innovative approaches are coming to the forefront to offer more potent and selective targeting of theoretically any protein of interest, including those deemed untargetable by the small molecules. These technologies include, among other: proximity-based approaches, including targeting chimeras (PROTAC/DUBTAC) and molecular glues, that bridge the protein of interest into close proximity to an E3 ligase or DUB for its degradation or stabilization [238,239]; tailored ubiquitin variants (UbVs) that bind with superior specificity to E3s and DUBs and can either activate or inhibit their actions [240]; and hydrophobic tagging, where synthetic hydrophobic ligands are used to trigger the unfolded mechanism of proteasome degradation of the target protein [241].

**Table 1 ijms-22-10800-t001:** Summary of the existing small molecules targeting the proteasome, E3 ligases, and DUBs. The total number of small molecules per family of enzymes as well as for each individual enzyme and subunits of the proteasome are listed. Annotations are made to quantify number of compounds on preclinical (PC) and clinical testing (C) or FDA approved. Due to the vast diversity of E3 and DUBs the list is updated and comprehensive, listing almost 250 compounds, but it is possible that it is not complete. The small molecules include small molecule inhibitors targeting the catalytic enzymatic domain, as well as those acting as antagonist or blockers of protein–protein interactions, based on published articles [16,242,243,244,245,246,247,248], and public databases.

Proteasome (Total = 39)		E3 Ligases (Total = 94)		DUBs (Total = 109)	
20S (total = 6)		RING-type (total = 70)		USPs (total = 77)	
20S	6 (FDA = 3, C = 3)	MDM2	25 (C = 13, PC = 8)	USP7	30 (PC = 18)
19S (total = 31)		SKP2	9 (C = 1, PC = 8)	unspecific-USP	27 (C-1, PC = 15)
USP14	14(PC = 4)	IAPs	5 (C = 4, PC = 1)	USP1	8 (C = 1, PC = 6)
unspecific-USP14	10(C = 2, PC = 4)	Unspecific- RING	5 (PC = 2)	USP2	3 (C = 1)
unspecific- RPN11	4(PC = 3)	Cul4-DCAF15	C = 4	USP30	3
RPN11 (PSMD14)	3(PC = 1)	MDMX	3 (PC = 2)	USP8	2
ATPase (total = 2)		RNF4	PC = 3	USP9X	PC = 1
p97	C = 2	XIAP	C = 2	USP19	1
		KEAP1	C = 2	USP20	1
		APC/C	PC = 2	USP28	1
		β-TrCP1	PC = 2	UCHs (total = 18)	
		TRAF6	PC = 2	UCHL1	14 (PC = 4)
		FBW7	PC = 2	UCHL3	3 (PC = 1)
		Met30	PC = 1	unspecific-UCH	1
		VHL	3	OTUs (total = 3)	
		HECT-type (total = 17)		TRABID	1
		E6AP	PC = 8	OTUB2	1
		HUWE1	PC = 2	Cezanne	1
		SMURF1/2	PC = 1	JAMMs (total = 3)	
		unspecific-HECT	PC = 1	CSN5	2 (PC = 1)
		WWP2	5	STAMBP(AMSH)	PC = 1
		RBR-type (total = 7)		Other DUBs (total = 8)	
		HOIP	7 (FDA = 1, PC = 3)	ADRM1(RPN13)	3 (PC = 1)
				SARS PLPro	3
				Ataxin	1
				Unspecific	1

FDA: U.S. Food and Drug Administration approved; C: in clinical studies; PC: in preclinical studies. DUB: deubiquitinase; HECT: Homologous to E6AP C-terminus ubiquitin ligases; RING: really interesting new gene ubiquitin ligases; RBR: RING-in-between-RING ubiquitin ligases; USPs: ubiquitin specific peptidases type of deubiquitinases; UCHs: Ubiquitin C-terminal hydrolase type of deubiquitinase; JAMMs: JAB1/MPN/MOV34 metalloenzymes type of deubiquitinase; OTUs: ovarian tumor proteases type of deubiquitinase; ADRM1(RPN13): adhesion regulating molecule 1; APC/C: anaphase-promoting complex/cyclosome; CSN5: COP9 signalosome complex subunit 5; Cul4-DCAF15: CUL4 and DDB1 associated factor 15; E6AP: E6-associated protein; FBW7: F-Box and WD repeat domain containing 7:; HOIP: HOIL-1L-interacting protein; HUWE1: HECT, UBA and WWE domain containing E3 ubiquitin protein ligase 1; IAP: inhibitors of apoptosis protein; Keap1: Kelch-like ECH associated protein 1; MDM2: mouse double minute 2 homolog; MDMX: murine double minute X; Met30: F-box protein Met30; OTUB2: OTU domain-containing ubiquitin aldehyde-binding protein 2; RNF4: ring finger protein 4; RPN11: 26S proteasome non-ATPase subunit Rpn11; SARS PLPro: severe acute respiratory syndrome papain-like protease; SKP2: S-phase kinase associated protein 2; SMURF1/2: SMAD ubiquitylation regulatory factor 1/2; STAMBP(AMSH): signal transducing adaptor molecule binding protein; TRABID: TRAF-binding domain; TRAF6: TNF receptor associated factor 6; UCHL1/3: ubiquitin carboxy-terminal hydrolase L 1/3; USP1/2/7/8/19/20/28/30/9X: Ubiquitin specific protease (USP) 1/2/7/8/19/20/28/30/9 X-Linked, respectively; USP14: ubiquitin carboxyl-terminal hydrolase 14; VHL: von hippel-lindau tumor suppressor; WWP2: WW domain containing E3 ubiquitin protein ligase 2; XIAP: X-linked inhibitor of apoptosis protein; β-TrCP1: beta-transducin repeat containing E3 ubiquitin protein ligase 1.

Most of the ubiquitin-based small molecule compounds have been designed to target essential ubiquitin pathways in cancer cells. With the advent of T-cell-based cancer immunotherapy, the exploration of targeted inhibition of selective E3s and DUBs for enhancing T-cell anti-tumor responses has commenced. Among those, Cbl-b, the key intracellular checkpoint T-cell inhibitor holds enormous potential. As discussed here, Cbl-b has pleiotropic anti-tumor effects in T cells, and its absence promotes T-cell anti-tumor responses at numerous levels: releasing CD28-dependence for T-cell activation and making them resistant to Tregs, impairing the inhibitory function of PD-1 and CTLA-4, as well as preventing T-cell anergy and exhaustion. Tumor studies on *Cbl-b*-catalytic-deficient *knockin* mice confirmed that the targeted inactivation of Cbl-b E3 ligase activity can lead to the same potent anti-tumor responses than those seen in *Cbl-b knockout* mice; again, without detectable autoimmune toxicities [129,138].

Unfortunately, selective small molecules inhibitors targeting Cbl-b, a RING-type of E3, have not yet been possible. Nevertheless, scientists have successfully down-regulated Cbl-b levels in T cells using RNA interference, which has yielded excellent in vitro and preclinical results in different adoptive T-cell transfer tumor models [139,140,249,250], and being well tolerated by cancer patients [251,252], it has now moved on to clinical trials (NCT02166255 and NCT03087591). Small peptides and small molecule inhibitors have been able to efficiently block in vivo Cbl-b interactions with a specific substrate [253] or inhibit directly the function of the Cbl-b relevant substrate [129]; yet, these approaches depend on substrate identification, which is often challenging. CRISPR/Cas9 technology has been successfully used in mice for targeted Cbl-b depletion in order to efficiently reconstitute the effector functions of exhausted tumor-infiltrating T cells [131]. Given that ubiquitin variants have been successfully developed to specifically inhibit the activated from of c-Cbl by blocking the E2-ubiquitin-binding site [254], it is likely that a similar approach could be utilized to inhibit Cbl-b. In addition, small molecule inhibitors for other key E3 ligases in T-cells responses are now available (Table 1); some of which are promising for use in immunotherapy. Small molecules against IAPs, MDM2 and deubiquitinase USP7 have already being tested preclinically and were shown to enhance anti-tumor immunity in vivo when used alone or in combinational therapies, at least partially, due to the intrinsic role of these E3 ligases in T-cell activities [255,256,257,258].

Ubiquitination-dependent pathways regulating PD-L1 degradation in cancer cells are also being tested for enhancing checkpoint therapy. For instance, a competitive palmitoylation inhibitor (CPP-S1) was developed to prevent PD-L1 palmitoylation and in turn, promote PD-L1 ubiquitin-mediated degradation [230]. The EGFR signaling pathways controlling PD-L1 ubiquitination and degradation have also been effectively blocked in vivo by different means, including EGFR small molecule inhibitors, copper chelators, and a small molecule (berberine) targeting the deubiquitinase CSN5; in all cases significantly enhancing T-cell antitumor responses [218,232].

Proximity-based therapeutic modalities, including PROTAC or molecular glue technologies, are also being rapidly developed to target for proteasomal degradation of several specific tumor proteins; there are currently over a dozen of clinical trials for validating the therapeutic potential of these ubiquitination-based technologies [259]. For immunotherapy, a recent study designed an antibody-based PROTACs (AbTACs) that was able to induce, without large cellular perturbations, PD-L1 ubiquitin-dependent lysosomal degradation by recruiting to PD-L1 the E3 ligase RNF43 [260]. These innovative methods are also being tested for next-generation CAR-T therapies. For this, the CAR is tagged with intracellular domains that can bind molecular glues (lenalidomide-based) or PROTAC (bromodomain-based) in order to provide reversible off CAR switches [261,262]. Upon addition of the protein degrader, the CAR protein is ubiquitinated and degraded at the proteasome and CAR protein expression was restored after removal of the protein degrader.

## 6. Concluding Remarks and Perspectives

The arrival and success of the newest protocols for immunotherapy, including checkpoint blockade and adoptive cell transfer of TCR-specific T cells, has transformed the field of cancer therapeutics. We have entered a period where experimental evidence and molecular mechanistic insight sets the basis for targeted treatments that are not only possible but often advantageous compared to broad-spectrum conventional therapies. To further progress down this road of precision medicine and offer more potent and less toxic treatments, we need to keep on widening the universe of relevant targets that goes beyond the cell surface and into the core of intricate intracellular networks.

In this review, we have covered in depth the most notable functions of ubiquitination in T cells, which fine-tune at multiple levels, T-cell activation and inhibition, tightly controlling the fate and function of the key signaling cascades downstream the TCR as well as key co-stimulating and co-inhibitory receptors. More importantly, we highlighted the genetic studies and preclinical data that demonstrate that modulation of several ubiquitin-dependent pathways, and in particular the ubiquitin-dependent enzymes involved, can unleash strong, long-lasting, and targeted anti-tumor responses. Fortunately, the ubiquitin field has also now made tremendous progress in developing ingenious tools to either specifically target virtually all ubiquitin-dependent enzymes or leverage them for targeting substrate proteins of interest, including oncoproteins. We look ahead confident that modulation of these essential ubiquitin dependent pathways could soon be a viable alternative to improve targeted cancer immunotherapy.

## Figures and Tables

**Figure 1 ijms-22-10800-f001:**
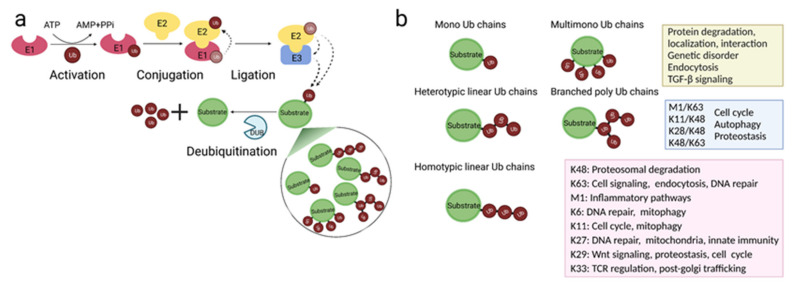
Mechanisms and functions of the ubiquitin system. (**a**) Enzymatic steps and enzymes involved in protein ubiquitination, a reversible and versatile post-translational modification. (**b**) Types of ubiquitin chains and their, thus far, identified cellular functions. Each ubiquitin chain has a different topology and is presumed to have distinct functions. The lysine 48 (K48)- and lysine 63 (K63)-linked polyubiquitin chains are abundant and well-studied; for the other types of chains, many known as atypical chains, cellular functions are starting to be revealed. Ub: ubiquitin; E1: ubiquitin-activating enzyme; E2: ubiquitin-conjugating enzyme; E3: ubiquitin ligase; DUB: deubiquitinase; AMP: adenosine monophosphate; ATP: adenosine triphosphate; PPi: inorganic pyrophosphate; K: Lysine; M: methionine; TGF-β: transforming growth factor-β. Illustration created using BioRender (biorender.com).

**Figure 2 ijms-22-10800-f002:**
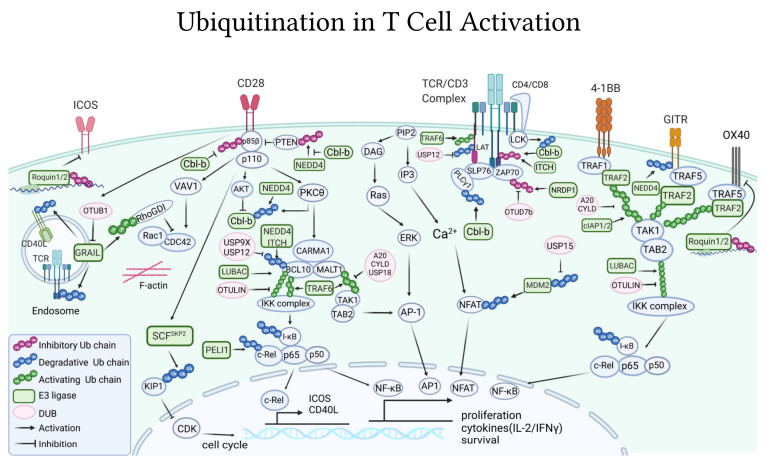
Ubiquitination in key T-cell activation pathways. T cells are activated upon antigen recognition by TCR complex (signal 1) and simultaneous engagement of co-stimulatory receptor such as CD28, 4-1BB, and ICOS (signal 2). These dual pathways integrate to trigger numerous signaling pathways that converge in the activation of NF-κB, NFAT, and AP-1 transcription factors that drive the genetic program for T-cell survival, proliferation, differentiation, and cytokine production. Various ubiquitin E3 ligases and deubiquitinases regulate these essential cascades using different ubiquitin chains, some of which activate (green chains), inhibit (purple chains), or degrade, via the lysosome or proteosome, (blue chains) the protein substrates. Ub: ubiquitin; DUB: deubiquitinase; AKT: serine/threonine-protein kinase/protein kinase B; AP-1: activator protein 1; BCL10: B cell lymphoma/leukemia 10; CARMA1: caspase recruitment domain-containing membrane-associated guanylate kinase protein-1; Cbl-b: casitas B-Lineage Lymphoma proto-oncogene B; CD40L: CD40 antigen ligand; CDC42: Cell division cycle 42; CDK: cyclin-dependent kinase; cIAP1/2: cellular inhibitor of apoptosis protein 1/2; CYLD: CYLD lysine 63 deubiquitinase; DAG: diacylglycerol; ERK: extracellular signal-regulated kinases; GITR: glucocorticoid-induced TNFR-related protein; GRAIL: gene related to anergy in lymphocytes protein; ICOS: inducible T-cell costimulator; IFNγ: interferon gamma; IKK complex: IκB kinase complex; IL2: interleukin 2; IP3: inositol trisphosphate; ITCH: itchy E3 ubiquitin protein ligase; I-κB: inhibitor of nuclear factor kappa B; KIP1: cyclin-dependent kinase inhibitor 1B; LAT: linker for activation of T cells; LCK: lymphocyte-specific protein tyrosine kinase; LUBAC: linear ubiquitin chain assembly complex; MALT1: mucosa-associated lymphatic tissue 1; MDM2: mouse double minute 2 homolog; NEDD4: neural precursor cell expressed developmentally down-regulated protein 4; NFAT: nuclear factor of activated T-cells; NF-κB: nuclear factor kappa-light-chain-enhancer of activated B cells; NRDP1: neuregulin receptor degradation protein-1 (also RNF41, ring finger protein 41); OTUB1: OTU domain-containing ubiquitin aldehyde-binding protein 1; OTUD7b: OTU deubiquitinase 7B; OTULIN: OTU deubiquitinase with linear linkage specificity; OX40: tumor necrosis factor receptor superfamily member 4 (also TNFRSF4); PIP2: phosphatidylinositol 4,5-bisphosphate; PKCθ: protein kinase C Theta; PLCγ1: phospholipase C-γ1; PTEN: phosphatase and tensin homolog; Rac1: rac family small GTPase 1; RhoGDI: Rho GDP-dissociation inhibitor; Roquin1/2: ring finger and CCCH-type zinc dinger somains 1/2; SCFSKP2: Skp1-Cullin-1-F-box (SCF) Cullin-Ring E3 ubiquitin ligase complex containing S-phase kinase associated protein 2 (SKP2); SLP76: lymphocyte cytosolic protein 2; TAB2: TGF-beta activated kinase 1 Binding Protein 2; TAK1: TGF-beta activated kinase 1 (also MAP3K7); TCR: T-cell receptor; TRAF1/2/5/6: TNF receptor associated factor 1/2/5/6; USP12/15/18/9X: ubiquitin specific peptidase 12/15/18/9 X-Linked, respectively; VAV1: vav guanine nucleotide exchange factor 1; ZAP70: ζ-chain associated protein 70. Illustration created using BioRender (biorender.com).

**Figure 3 ijms-22-10800-f003:**
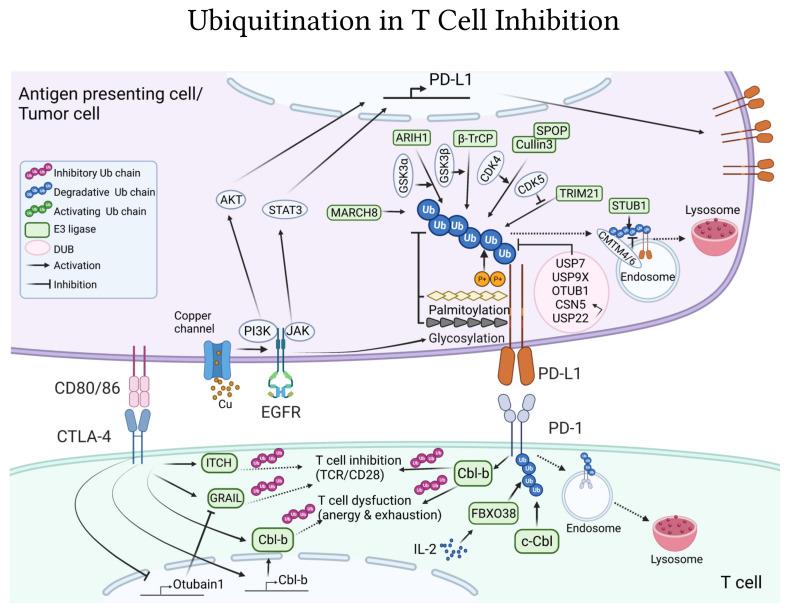
Ubiquitination in the CTLA-4 and PD-1/PD-L-1 checkpoint inhibitory receptors. Upon interaction with their cognate ligands on the cell surface of APCs or tumor cells, T-cell co-inhibitory receptors such as CTLA-4 and PD-1 impair T-cell activation to shut down the immune response and permit tissue repair. The ubiquitin system controls the function and stability of these receptors through degradative (blue) and non-degradative chains (purple), significantly impacting the outcome of these signaling events. Ub: ubiquitin; P+ (orange circle): phosphorylated residues; ARIH1: ariadne RBR E3 ubiquitin protein ligase 1; AKT: serine/threonine-protein kinase/ protein kinase B; Cbl-b: casitas B-lineage lymphoma proto-oncogene B; c-Cbl: casitas B-lineage lymphoma proto-oncogene C; CDK4: cyclin-dependent kinase 4; CDK5: cyclin-dependent kinase 5; CMTM4/6: CKLF-like MARVEL transmembrane domain containing 4/6; CSN5: COP9 signalosome complex subunit 5; CTLA-4: cytotoxic T-lymphocyte-associated protein 4; Cu: Copper ions; EGFR: epidermal growth factor receptor; FBXO38: F-box protein 38; GRAIL: gene related to anergy in lymphocytes; GSK3α: glycogen synthase kinases α; GSK3β: glycogen synthase kinases β; IL2: interleukin 2; ITCH: itchy E3 ubiquitin protein ligase; JAK: janus kinase; MARCH8: membrane-associated ring finger 8; OTUB1: OTU domain-containing ubiquitin aldehyde-binding protein 1; PD-1: programmed cell death protein 1; PD-L1: programmed death-ligand 1; PI3K: phosphoinositide 3-kinase; SPOP: speckle type BTB/POZ protein; STAT3: signal transducer and activator of transcription 3; STUB1: STIP1 homology and U-box containing protein 1; TRIM21: tripartite motif containing 21; USP7/22/9X: ubiquitin specific peptidase (USP) 7/22/9X-Linked, respectively; β-TrCP: beta-transducin repeat containing E3 ubiquitin protein ligase. Illustration created using BioRender (biorender.com).

## Data Availability

Not applicable.

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
