# Peer review of "Ubiquitination in T-Cell Activation and Checkpoint Inhibition: New Avenues for Targeted Cancer Immunotherapy"

_ijms, 2021, doi:10.3390/ijms221910800_

Round 1

Reviewer 1 Report

Reviewer comments and suggestions

The current review discussed the molecular and functional roles of ubiquitination in key T cell activation and checkpoint inhibitory pathways to highlight the vast possibilities that targeting ubiquitination offer for advancing T-cell-based immunotherapies.

Thus, a deeper understanding of the molecular mechanisms of T cell activation and inhibition is much needed to rationally expand targets and possibilities to improve immunotherapies.

Decision: Minor comments

Below are the comments for this paper to be incorporated in the revised version of the manuscript. 

  1. Line 30-31 Please explore the sentence “Unfortunately, the same anti-tumor immune response often applies selective pressure for the tumor to escape immunity”
  2. Line 42 toxicities associated such as 
  3. Line 70-72 sentences need to be explored
  4. Line 80-81 Please explain comprehensibly reference 25
  5. Line 175-176 explore the cited references
  6. Line 225 mouse model what does it mean here
  7. Line 248, 225 421 typo error please modify
  8. I would suggest the author to indicate figure numbers in the text in this way to easily follow up the paper
  9. Line 309, please check the representation Nrdp1 NRDP1
  10. Line 351-352 it seems that sentence was not complete
  11. pd1 pathway needed to discuss in 4.1 sections similar to CART
  12. A long sentence at the end of the conclusion was not needed. It should be short and meaningful. 

Reviewer 2 Report

Summary

This review described a general outline of mechanisms of T cell activation and inhibition, and the complex role(s) of multiple components of the ubiquitinylation system in controlling T cell signal integration and propagation.  The review is generally very well structured, well written and extensively referenced.  This review was an interesting and extremely informative piece of work. However, some statements lacked appropriate references or were incompletely described and explored, and the review contained many small typographical and grammatical errors. These are listed in detail below, and should be addressed prior to publication.  Overall, congratulations to the authors on a comprehensive and valuable piece of work. 

General Comments

Line 79: “SUMO”(ylation) and “NEDD”(ylation) should be fully defined before abbreviating, or alternatively written in full in a list of abbreviations, as these terms may be unfamiliar to many readers.

Line 120:  the authors’ statement that “the human genome codes for 2 E1s, around 40 E2s, 700 E3s and almost 100 DUBs” should have reference to either literature sources or a public database. 

Line 153:  the authors should specify that a “second signal is needed to activate naïve T cells” rather than “T cells”, as this is not a stringent requirement for effector or memory T cell reactivation.   

Line 221-222:  the authors’ statement that “T cells experiencing repeated antigen-driven proliferation can enter a state of senescence and become cell cycle arrested” should be referenced. 

Lines 291-292:  the authors state that absence of cbl-b “restores in mice T cell anergy and exhaustion” – this is written in a confusing manner, as I assume the authors mean that absence of cbl-b alleviates or prevents anergy and exhaustion?  Could the authors please reframe this statement and possibly provide detail as to the experimental systems used (i.e. knockout mice, ex vivo CRISPR, transient silencing etc). 

Section 4.1.1:  Is the silencing or loss of cbl-b associated with autoimmune symptoms in mice?  As the authors propose this as an important therapeutic target and provide good evidence of its role in controlling T cell activation, discussion of this possibility should be included in this section.   Similarly, what (if anything) is known the long-term consequence of a loss of ubiquitin ligase function?  Do T cells become exhausted from continued unopposed signalling?

Line 312:  The authors make reference to several USPs throughout the review, but do not provide a full name (i.e. Ubiquitin specific peptidase) at any point – this should be done once at the first use.      

Line 440:  The authors state that “The loss of TRAF2 in mice presents delayed activity under agonist antibody-based 4-1BB immunotherapy in mice” – could the authors please clarify and specify what this delayed activity is?  The authors could also consider removing the second use of “in mice” at the end of the sentence as it is repetitive.

Line 443: The authors state that “In primary human cytotoxic CD8+ T cells, the knockdown of A20 or CYLD promoted 4-1BB co-stimulation” – again this is vague, could the authors please clarify what was being measured and what specifically was being promoted (T cell proliferation, cytokine production etc).

Figures 2 and 3:  The authors might consider removing the large blue header boxes from these diagrams, but this is only an aesthetic comment.

Figure 2:  Text within the nucleus needs to be changed (“cytokines” should be capitalised for consistency, and “Cd40l” should be changed to “CD40L”).

Figures 1-3: The authors have not attributed the production of the figures or noted which image software was used. 

Line 492:  The authors state that “Further, these three E3 ligases also play essential immunosuppressing roles by regulating Tregs” but do not provide any further mechanistic detail or evidence as to how this is occurring and what the consequences are.  This should be included. 

Line 594: “wisely interfering” is an unusual choice of words, perhaps consider changing to “rationally interfering” or similar.

Line 652-653:  The authors state that Cbl-b down-regulation has “now moved on to clinical trials” – here they should specify NCT designations (or similar) for these trials as evidence.

Line 696: “notorious functions” is an unusual choice of words, perhaps consider changing to “notable functions” or similar.

Typographical and other errors.

Abstract line 19:  Replace “offer” with “offers”.

Line 44: Replace “mechanism” with “mechanisms”.

Line 51: Replace “revised” with “reviewed”.

Figure 1b: The yellow box contain the word “ecndocytosis”, rather than endocytosis. 

Line 244: Replace “CTLA-4 target” with “targeted” or “targeting”.

Line 248: Remove the underscore from “PD-1/_PD-L1”.

Line 255: Remove the rectangular box from “CD3ζζITAMs”.

Line 327: The authors have used “Usp” but in all other cases used “USP”.

Line 334: Replace “alter T cells” with “alter T cell function” or similar.

Line 356: “Despite it is long known that” should be removed and replaced with “Although it has long been known that” or similar.

Line 374:  Add “downstream of co-stimulatory receptors”.

Line 409: Replace “ubiquitylation” with “ubiquitinylation”.

Line 415: Replace “leads” with “lead”.

Line 421: Replace “NF-B” with “NF-kB”.

Line 453: Replace “functions” with “function”.

Line 455: Replace “functions” with “function”.

Line 458: Replace “ubiquitinate it” with “ubiquitinates it”.

Line 459: Remove “are”.

Line 460: Replace “decreases” with “decrease”.

Line 489-490: Replace “downstream CTLA-4” with “downstream of CTLA-4”.

Line 526: Replace “T cell” with “T cells”.

Line 542: Replace “phosphorylate” with “phosphorylates”.

Line 584: Add a space between “[208].It”.

Line 587: Replace “OTUB1, USP9X, USP7.” with “ OTIB1, USP9X and USP7”.

Line 636: Add a space between “DUBs(total)” in table header.  The last row in the central column also reads “HOIP 7 *@7(FDA=1, PC=3)” – this should be amended.

Line 694: Replace “intricated” with “intricate”.
